# Corneal Descemetocele Management with Multi-Layer Amniotic Membrane Transplantation in an Ocular Graft-versus-Host Disease Case

**DOI:** 10.3390/medicina59101733

**Published:** 2023-09-27

**Authors:** Yunjiao He, Hiufong Wong, Jianjun Gu, Lixia Lin

**Affiliations:** 1State Key Laboratory of Ophthalmology, Zhongshan Ophthalmic Center, Guangdong Provincial Key Laboratory of Ophthalmology and Visual Science, Guangdong Provincial Clinical Research Center for Ocular Diseases, Sun Yat-sen University, Guangzhou 510060, China; hyjdhyy@163.com (Y.H.); wong63@mail2.sysu.edu.cn (H.W.); gujj@mail.sysu.edu.cn (J.G.); 2Department of Laboratory, Dehong People’s Hospital, Dehong 678400, China

**Keywords:** ocular graft-versus-host disease (oGVHD), corneal descemetocele, sterile corneal melting, amniotic membrane transplantation, case report

## Abstract

*Background*: Chronic ocular graft-versus-host disease (oGVHD) is a common ocular complication following allogeneic hematopoietic stem cell transplantation (allo-HSCT), characterized by progressive inflammation of the ocular surface and refractory dry eye. In severe cases, sterile corneal perforation can occur, which poses a significant challenge, due to the low survival rate of grafts after corneal transplantation. *Case Presentation*: A 47-year-old female presented to our hospital with persistent dryness, foreign body sensation, and blurred vision in her left eye. Diagnosis of graft-versus-host disease with corneal descemetocele in the left eye was made after detailed history review and thorough examination. Multi-layer amniotic membrane transplantation was performed in the affected eye, resulting in amelioration of the patient’s symptoms. This amelioration of symptoms provided the patient with a level of comfort that permitted additional time while awaiting corneal transplantation. *Conclusions*: We report a successful case of multi-layer amniotic membrane transplantation for the management of corneal descemetocele following allo-HSCT.

## 1. Background

Allogeneic hematopoietic stem cell transplantation (allo-HSCT) is an effective treatment for a variety of hematologic disorders [1].With advancements in technology, the survival rates and durations of survival for allo-HSCT patients have improved. Graft-versus-host disease (GVHD) is a severe and potentially life-threatening complication that may arise after allo-HSCT. Chronic ocular graft-versus-host disease (oGVHD) is the most common persistent ocular complication, characterized by progressive inflammation of the ocular surface, refractory dry eye, superficial punctate keratitis, and peripheral corneal neovascularization. Severe complications include sterile corneal lytic perforation and symblepharon [2]. The management of oGVHD, particularly in patients with corneal melting, is a significant challenge. In this report, we describe a case in which multilayer amniotic membrane transplantation was used to treat corneal descemetocele after allo-HSCT.

## 2. Case Presentation

A 47-year-old female patient presented with persistent dryness, foreign body sensation, and blurred vision in her left eye, which had begun several weeks after her bone marrow transplantation one and a half years ago. She had been using various types of lubricants as often as she needed (such as polyethylene glycol eye drops and hyaluronic acid eye drops) since the onset of the dry eye. Then, she was referred to our hospital, as her symptoms had significantly worsened along with unbearable pain over the previous week. She had previously been diagnosed with hemophagocytic syndrome and alpha thalassemia and had undergone a 12/12 identical allo-HSCT a year and a half prior. The patient had been diagnosed with skin GVHD, oral GVHD, and pulmonary GVHD before she was referred to our hospital. To manage her systemic condition, oral immunosuppression and corticosteroids (tacrolimus and prednisone) were prescribed, and the drug dose was adjusted accordingly. A dosage of 0.04 mg/kg of tacrolimus was administered for 15 days prior to transitioning to a dosage of 1 mg, then adjusted to 0.5 mg as a maintenance dose. Methylprednisolone was started with a daily dose of 1000 mg for a duration of 3 days. Subsequently, prednisone was administered at a dose ranging from 0.5 to 1 mg/kg/day and then gradually tapered to maintenance dose of 10 mg.

Upon examination, the patient’s right eye (OD) had a best-corrected visual acuity (BCVA) of 20/25, whereas the left eye (OS) had a BCVA of hand movement. The non-contact intraocular pressure (NCT) of the OD was 18 mmHg, whereas that of the OS was not measured. The Schirmer I test without anesthesia showed 8 mm/5 min for the OD and 2 mm/5 min for the OS. Slit-lamp examination revealed mild conjunctival hyperemia and punctate epithelial erosion in the OD (Figure 1A,B), whereas the OS showed moderate ciliary hyperemia without edema and an oval, 8 mm × 9 mm, gray-white, and cloudy stromal ulcer located just below the center of the cornea. The ulcer had exposed Descemet’s membrane, which measured 2 mm × 3 mm in size at the center of the corneal ulcer (Figure 1C,D). Anterior segment optical coherence tomography (AS-OCT) identified the thinnest corneal point inferiorly to the center in the OS, measuring 0.06 mm. B-scan ultrasound did not detect any abnormal echoes in both eyes (OU).

The diagnosis was confirmed as ocular graft-versus-host disease with corneal descemetocele in the left eye based on the patient’s history and clinical findings. The affected eye underwent multi-layer amniotic membrane transplantation, and the patient continued to receive oral immunosuppression and corticosteroid therapy of maintenance dose after the surgery.

The surgery was performed using an operating microscope under general anesthesia. The surgical procedure was performed as follows: initially, the necrotic tissue and exudates on the surface of the corneal ulcer were scraped using cotton swabs (Figure 1A). An amniotic membrane was gently folded to 10 layers under the microscope using smooth forceps and placed in the ulcer area, serving as a scaffold of cornea stroma (Figure 2B). Subsequently, a single layer of amniotic membrane was placed stromal side down on top of the 10-layered membrane, as shown in Figure 2C. Interrupted 10-nylon sutures were used to secure the amniotic membrane in place, as shown in Figure 2D. Intraoperative anterior segment optical coherence tomography (AS-OCT) was performed to confirm that the corneal arc and thickness were appropriately restored after the amniotic membrane placement in the cornea, as shown in Figure 3. Finally, a corneal bandage contact lens was applied. After the surgery, tacrolimus eye drops (0.1%) were prescribed twice a day for long-term use; fluorometholone eye drops (0.1%) were prescribed twice a day for a month; ofloxacin eye drops (0.3%) were prescribed four times a day for a month; and polyethylene glycol eye drops (0.4%) were prescribed four times a day for long-term use.

After three months, most of the sutures had become loose and were removed. The patient’s symptoms of dryness and pain gradually improved after the surgery. The BCVA of the OS remained at hand movement and the IOP was estimated to be normal using the finger tension technique. The amniotic membrane had gradually integrated with the cornea (Figure 4A), and the corneal thickness was restored (Figure 4B). Neovascularization was observed in the cornea under slit lamp biomicroscopy.

## 3. Discussion

Chronic oGVHD is a common complication in allo-HSCT patients, affecting 30% to 60% of patients who undergo allo-HSCT and 60% to 90% of patients who develop GVHD. Ocular GVHD has a significant impact on the patients’ quality of life. However, the pathological mechanisms that lead to oGVHD are not fully understood, despite extensive research [3,4,5,6,7,8]. It is currently believed that the main mechanism of oGVHD involves a complex immune response mediated by donor-derived CD4+ and CD8+ T cells. These cells recognize and attack host tissue histocompatibility antigens, resulting in significant activation of the immune system and progressive fibrosis through multiple inflammatory mechanisms [9,10,11]. During the initial phase, T cells infiltrate the glandular ductal epithelial cells of the lacrimal gland and initiate an immune–inflammatory cascade. This cascade induces apoptosis, recruits immune cells, and produces inflammatory cytokines and chemokines, which ultimately leads to fibrosis of the lacrimal duct and stroma. This fibrosis causes a marked increase in the number of CD34+ fibroblasts and lymphoproliferation, resulting in decreased tear secretion [12,13,14]. The depletion of tears and immune cell infiltration can lead to the loss of conjunctival goblet cells and corneal nerve fibers. Inflammatory and fibrotic changes in the eyelids and meibomian glands can also exacerbate ocular surface damage [15,16,17,18]. Therefore, the main manifestation of oGVHD is refractory dry eye caused by a combination of fibrosis of the lacrimal and meibomian glands, reduction in conjunctival goblet cells, immune keratoconjunctivitis, and damage to the ocular surface microenvironment. Severe complications include corneal neovascularization, corneal melting, and corneal perforation [19,20,21,22].

Currently, there is no specific treatment for oGVHD, and clinical management primarily involves a multidisciplinary approach to safeguard the ocular surface, alleviate symptoms, and prevent severe complications. Local pharmacological management focuses on anti-inflammatory therapy [9], with T-cell-targeted drugs like tacrolimus and cyclosporine eye drops demonstrating some effectiveness in reducing ocular surface inflammation and improving related symptoms [23]. While corticosteroid eye drops also have anti-inflammatory effects, adverse reactions such as epithelial toxicity, corneal dissolution, and ocular hypertension should be closely monitored. However, the current anti-inflammatory drug therapy still has limited effectiveness, highlighting the need for further exploration of the inflammatory mechanisms in oGVHD to achieve better clinical outcomes [24]. Preservative-free ocular lubricants are first-line treatments for oGVHD, whereas topical secretagogues like ripasudil and diquafosol sodium eye drops can improve tear secretion. Autologous serum can also promote epithelial repair and may be considered if necessary [25]. Physical therapy aiming at reducing tear loss and improving meibomian gland dysfunction might be helpful, including punctal occlusion, moist chamber therapy, and eyelid hygiene and massage [26,27]. In cases of limbal stem cell deficiency, allogeneic limbal stem cell transplantation can be attempted [28].

However, even with aggressive treatment, severe cases of oGVHD may still lead to serious ocular complications, such as corneal melting and perforation. Surgical interventions, including tarsorrhaphy, amniotic membrane transplantation, conjunctival transplantation, and even corneal transplantation, may be necessary. Yue Xu et al. reported the treatment of seven cases (nine eyes) of chronic oGVHD-related corneal perforation, which showed that despite aggressive treatments like conjunctival flap coverage and penetrating keratoplasty, most patients (seven eyes) still had a final visual acuity below 0.02 [29]. This is due to the presence of refractory inflammation and intractable dry eye in oGVHD patients, making corneal transplantation a high-risk procedure with a low postoperative graft survival rate and frequent complications such as graft rejection, dissolution, and infection [30,31]. As a result, the occurrence of corneal perforation significantly increases the difficulty in treatment and worsens the prognosis. Some researchers suggested that preventing oGVHD-related corneal perforation should be a priority to avoid the need for lamellar or penetrating keratoplasty [32]. The patient in our case had been diagnosed with systemic multi-organ chronic GVHD, along with refractory ocular surface inflammation as such; corneal transplantation at this time should be at high risk for graft failure and might even lead to secondary graft melting. All things considered, after fully communicating with the patient, we decided that the timing of corneal transplantation had not come yet. Therefore, we performed the amniotic membrane transplantation to prevent corneal perforation and avoid the potential need for emergency keratoplasty. Multilayer amniotic membrane transplantation was performed knowing the visual acuity might not improve postoperatively, and secondary keratoplasty will be optional in the future when her GVHD condition improved.

Amniotic membrane is a non-vascularized fetal membrane collected from placental tissue and is widely used in ophthalmology. Due to the absence of HLA-A, HLA-B, and HLA-DR antigens, it is well-tolerated after transplantation and can serve as a cellular growth scaffold. It also contains various anti-inflammatory and growth factors that can inhibit inflammation and promote epithelial and stromal healing [33]. Amniotic membrane transplantation is effective in treating various complex corneal epithelial healing disorders, ulcers, and perforations [33].

In recent years, several reports have shown that amniotic membrane transplantation in the acute phase of oGVHD can restore ocular surface integrity, improve visual prognosis, and avoid serious complications [34,35]. Anastasios and associates had reported a glueless and sutureless multi-layer amniotic membrane patching with a bandage contact lens, and such a simple procedure had successfully prevented the deterioration of corneal thinning [36]. Yet, the corneal ulcer in their case was relatively localized without corneal descemetocele, whereas our case had a larger and deeper ulcer, which might be more difficult for amniotic membrane to attach to the cornea without sutures. For oGVHD patients with corneal perforation, individual case reports have demonstrated successful closure of the corneal perforation with multilayered amniotic membrane transplantation, thus avoiding the need for corneal transplantation [37]. The advantage of using 10-layered amniotic membrane transplantation is that it can be used as a collagen scaffold to restore corneal structure with sufficient thickness by comparing it to 2–4 layers of amniotic membrane. Even though it was reported that corneas covered with amniotic membranes may pose a risk of overlooking signs of intraocular infection or inflammation due to a lack of transparency [38], it should be mentioned that corneal transparency, in this case, can only be restored through corneal transplantation, which might not be the optimum choice, as discussed above.

## 4. Conclusions

In this case, prompt surgical intervention was necessary for a patient presenting with severe symptoms, including unbearable pain and corneal melting as well as corneal descemetocele. Without intervention, there was a significant risk of corneal perforation with secondary endophthalmitis. As oGVHD-related corneal transplantation is a high-risk procedure with a low postoperative graft survival rate during the acute inflammatory phase, multilayered amniotic membrane transplantation was used instead. The amniotic membrane served as an effective corneal matrix scaffold, reconstructing the ocular surface structure and laying the foundation for restoring visual acuity with corneal transplantation in the future.

## Figures and Tables

**Figure 1 medicina-59-01733-f001:**
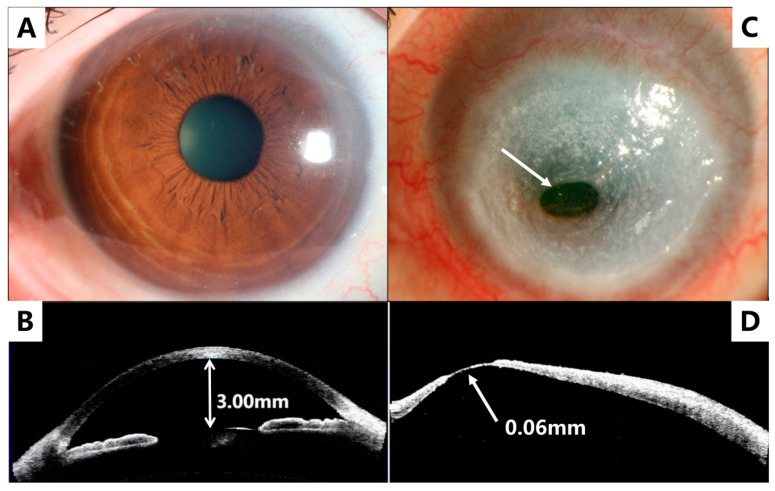
Preoperative anterior segment photography of right eye (**A**) appearing within normal limits and left eye (**C**) with corneal descemetocele (arrow) in the inferior central cornea within an oval cloudy stromal ulcer. Anterior segment optical coherence tomography (AS-OCT) of right eye (**B**) and left eye (**D**) with evidence of corneal thinning (arrow) consistent with the descemetocele observed in (**C**).

**Figure 2 medicina-59-01733-f002:**
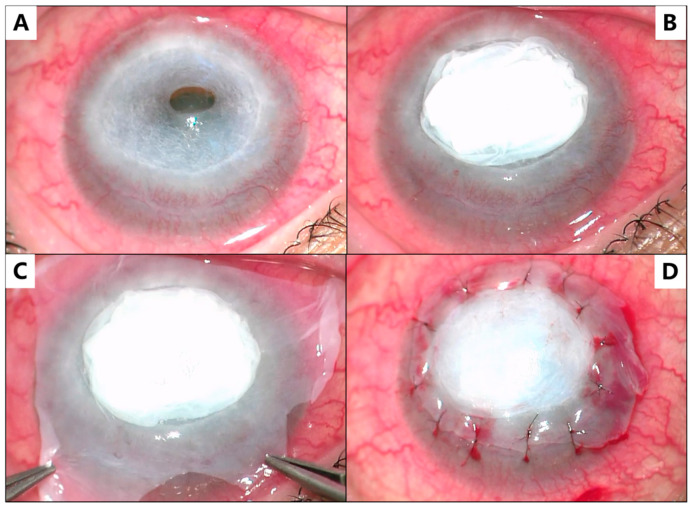
Intraoperative anterior segment photographs of the surgical procedure of multilayer amniotic membrane transplantation. (**A**) following debridement of the corneal ulcer; (**B**) the folded multilayered amniotic membrane positioned in the bed of the corneal ulcer; (**C**) a single layer of amniotic membrane covering the multi-layered amniotic membrane; and (**D**) following suturing with 10–0 nylon.

**Figure 3 medicina-59-01733-f003:**
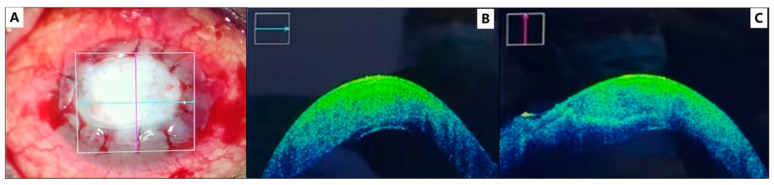
Anterior segment photography of the left eye right after the surgery (**A**). Intraoperative OCT after multilayer amniotic transplantation. Corneal thickness was restored to a more or less normal corneal thickness at the end of the operation, as presented in horizontal (**B**) and vertical (**C**) OCT scans.

**Figure 4 medicina-59-01733-f004:**
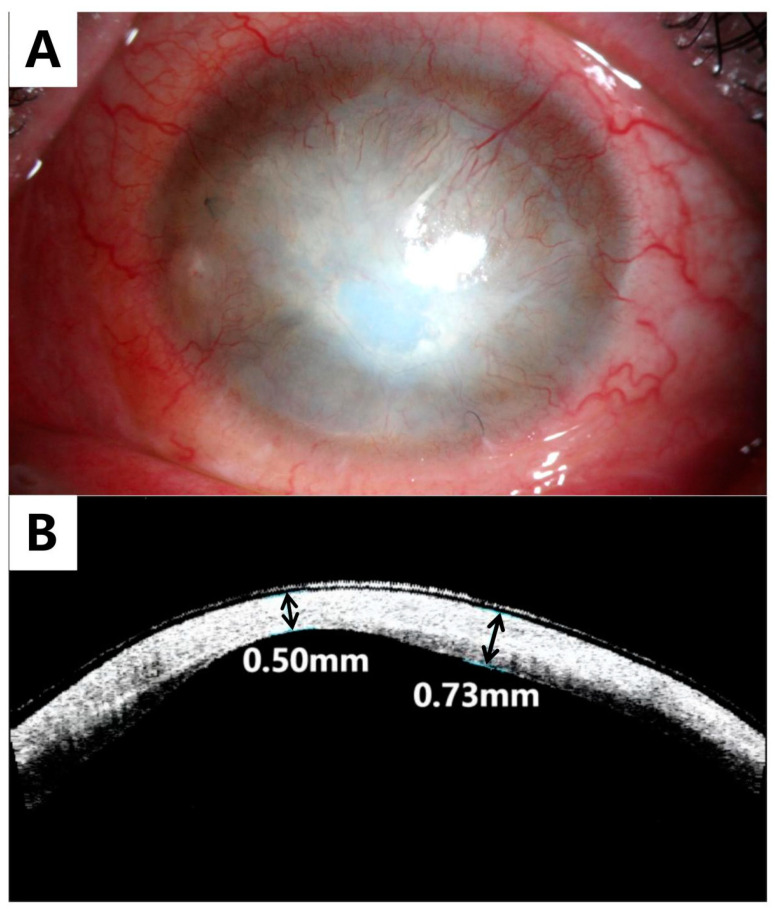
Anterior segment photography of the left eye three months after the surgery (**A**) and AS-OCT (**B**) demonstrating amniotic membrane integrated into the cornea with restoration of a more or less normal corneal thickness.

## Data Availability

Data sharing is not applicable to this article, as no datasets were generated or analyzed during the current study.

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
