# Peer review of "Corneal Descemetocele Management with Multi-Layer Amniotic Membrane Transplantation in an Ocular Graft-versus-Host Disease Case"

_medicina, 2023, doi:10.3390/medicina59101733_

Round 1

Reviewer 1 Report

He and colleagues reported successful management of a corneal ulcer complicated with descemetocele in the setting of chronic ocular graft versus host disease (oGVHD) using multilayer amniotic membrane. The case has been presented very well and the pre and postoperative corneal images are very useful. My comments for the authors are listed below:

1.       Case presentation: has the patient been receiving any treatment for dry eye disease (either lubricants or punctal occlusion, etc) prior to surgery?  
2.       Line 59: please explain what the Schirmer test results refer to. I assume they are with or without topical anesthesia (i.e., Schirmer I and II).
3.       Line 69: did the patient have any systemic signs/symptoms of chronic GVHD?
4.        Line 76-78: were the amniotic membranes cryopreserved? Also, please describe the orientation of the amniotic membranes. Were they placed stromal side up or down?
5.       Line 84: has the patient undergone punctal occlusion and/or lateral tarsorrhaphy? Also, how long after the surgery were the sutures removed?
6.       Figures 1 and 3: please add the OCT B-scan plane to the corneal photographs to show the location and orientation of the B-scans.
7.       I recommend the authors include and discuss other similar reports, for example, PMID: 34513346 and PMID: 34513346 in the discussion.
8.       Use of abbreviations can be more accurate and consistent. For example, in the abstract, allo-HSCT can be abbreviated in line 16 instead of line 29. Also, HSCT in line 118 can be replaced with allo-HSCT.
9.       Line 60: “punctate detachment of the corneal epithelium”, I suggest using “punctate epithelial erosion (PEE)” instead.
10.   Line 136: “immune-induced corneal conjunctivitis”, please consider revising this phrase.
11.   Line 154: “tear duct occlusion” may be replaced with “punctal occlusion”.
12.   Line 155: “corneal margin” should be replaced with “limbal”.

Minor English editing is required. The use of abbreviations and scientific terminologies can be more accurate and consistent.

Reviewer 2 Report

This case report highlights the crucial role of amniotic membrane transplantation in preventing corneal perforation in patients suffering from ocular graft-versus-host disease (GVHD). The manuscript is thoughtfully composed, providing valuable information for readers. The figures are well-presented and easily comprehensible. However, there are some minor concerns that warrant attention and addressing.

Line 15, Line 36, line 38, and Line 199. Hyphens are required to spell out the abbreviation "GVHD" as "graft-versus-host disease".

Line 48. How soon did the symptoms begin following hematopoietic stem cell transplantation? A more detailed description, including the duration in days or months, is required.

Line 51. Allogeneic hematopoietic stem cell transplantation has already been abbreviated as HSCT in the text. Please edit accordingly.

Line 59. Was the Schirmer test performed with or without anesthesia?  

Line 83-84. Description of the frequency and duration of topical eye drops is required. 

In the discussion, the author should consider more detailed discussion on reasons for not undergoing corneal transplantation first and Its considerations for improving the patient’s visual acuity. 

Figure legend 1-3. The results and explanations have already been described in the main text. The authors can shorten the description of legends without duplication from the main text.

GVHD should be spelled out at first appearance of text as graft-versus-host disease, key words and explanation for abbreviation.

Reviewer 3 Report

This manuscript reported a graft-versus-host disease (GVHD) case with corneal descemetocele managed with multilayer amniotic membrane transplantation (AMT). The details of systemic and topical treatment in her ocular GVHD before AMT were not clear. The dosage of systemic immunosuppression and corticosteroids (tacrolimus and methylprednisolone) should be better indicated for the control of chronic GVHD. Authors did not mention topical treatment course in this severe complication of ocular GVHD before AMT because proper systemic and topical treatment may prevent occurrence of corneal descemetocele compication. What kind of topical tacrolimus and artificial tears were used in postoperative medication? Please indicate concentration, forms (solution or ointment) and frequency. By the way, authors may explain the reasons for using 10-layered AMT because most multi-layer AMT usually uses 2-4 layers only in corneal descemetocele or perforation. Multi-layered AMT has been reported to use for corneal melting, perforation and corneal surface reconstruction with good results. Authors may discuss the pros and cons of using 10-layered AMT because too many layers amniotic membrane may obscure the visualization of the change of cornea and anterior segment or increase the risk of infection, which has been reported.

Round 2

Reviewer 1 Report

All my comments have been addressed satisfactorily in the revised manuscript. I have no further comments for the authors.

Author Response

Thank you for your letter and for the comments concerning our manuscript titled “Corneal descemetocele management with multi-layer amniotic membrane transplantation in an ocular graft-versus-host disease case.”

Yours sincerely,

Lixia Lin

State Key Laboratory of Ophthalmology, Zhongshan Ophthalmic Center, Sun Yat-sen University, Guangdong Provincial Key Laboratory of Ophthalmology and Visual Science, Guangdong Provincial Clinical Research Center for Ocular Diseases, Guangzhou 510060, China

Reviewer 3 Report

This paper showed a lesson "Severe corneal complication could occur if ocular GVHD did not receive aggressive systemic and topical treatment".  Initial ocular treatment in this case may be not good enough to prevent corneal melting in their local hospital.

Author Response

(The authors gave the same response as above.)
